# A Collaborative Virtual Walkthrough of Matera’s Sassi Using Photogrammetric Reconstruction and Hand Gesture Navigation

**DOI:** 10.3390/jimaging9040088

**Published:** 2023-04-21

**Authors:** Nicla Maria Notarangelo, Gilda Manfredi, Gabriele Gilio

**Affiliations:** 1School of Engineering, University of Basilicata, 85100 Potenza, Italy; 2Department of Mathematics, Computer Science, and Economics, University of Basilicata, 85100 Potenza, Italy

**Keywords:** virtual walkthrough, educational and cultural heritage applications, collaborative virtual learning environment, photogrammetry, hand gesture recognition (HGR), deep learning, human–computer interaction

## Abstract

The COVID-19 pandemic has underscored the need for real-time, collaborative virtual tools to support remote activities across various domains, including education and cultural heritage. Virtual walkthroughs provide a potent means of exploring, learning about, and interacting with historical sites worldwide. Nonetheless, creating realistic and user-friendly applications poses a significant challenge. This study investigates the potential of collaborative virtual walkthroughs as an educational tool for cultural heritage sites, with a focus on the Sassi of Matera, a UNESCO World Heritage Site in Italy. The virtual walkthrough application, developed using RealityCapture and Unreal Engine, leveraged photogrammetric reconstruction and deep learning-based hand gesture recognition to offer an immersive and accessible experience, allowing users to interact with the virtual environment using intuitive gestures. A test with 36 participants resulted in positive feedback regarding the application’s effectiveness, intuitiveness, and user-friendliness. The findings suggest that virtual walkthroughs can provide precise representations of complex historical locations, promoting tangible and intangible aspects of heritage. Future work should focus on expanding the reconstructed site, enhancing the performance, and assessing the impact on learning outcomes. Overall, this study highlights the potential of virtual walkthrough applications as a valuable resource for architecture, cultural heritage, and environmental education.

## 1. Introduction

The COVID-19 pandemic has wrought an unprecedented and far-reaching global impact, prompting governments worldwide to implement sweeping measures to contain the spread of the virus. These measures, including school and workplace closures, gathering and travel restrictions, and stay-at-home orders, have drastically altered people’s mobility and lifestyle patterns [1]. Such changes in people’s behavior have brought abruptly to the forefront the need for tools that enable users in different locations to interact and work together in a shared virtual space in real time. In particular, educational institutions have resorted to remote learning to deliver educational content and facilitate online classrooms, allowing students to collaborate and access learning materials remotely.

In this sense, collaborative eXtended Reality (XR) technologies, integrating Virtual Reality (VR), Augmented Reality (AR), and Mixed Reality (MR), have gained increasing popularity in the educational field as they largely proved their potential to improve learning quality and training performance [2] by providing immersive experiences that simulate real-world environments. Innovative educational methods, including serious games [3,4], and digital technologies such as digital survey, VR, and AR, have also greatly contributed to the virtual reconstruction of historical sites [5,6] and the dissemination of cultural heritage [7], intended as tangible culture, intangible culture, and natural heritage.

Advanced technologies and techniques in data acquisition, processing, representation, and analysis can improve three-dimensional (3D) digitization and comprehensive recording of valuable architectural and cultural sites. Innovative solutions like photogrammetry-based virtual reconstruction can tackle the issues of morphological and material complexity commonly encountered in cultural heritage sites [8] while enhancing new digital content management, representation, and reproduction [9,10]. Photogrammetry offers a straightforward, low-cost, and widely available method for obtaining digital replicas of real-world objects, enabling instructional designers to incorporate visually appealing stimuli in their learning content. Photorealistic 3D models can be particularly useful in preservation and archeology scenarios where precise reproduction of physical counterparts is essential. These models can enhance learning outcomes by providing a high level of realism and increasing accessibility [11] through digitization [12]. Despite the growing interest shown by the current research in the use of such technologies for cultural heritage and education [5,6], the challenge of providing seamless interaction with online, real-time visualization of reality-based 3D surveying and modeling of heritage sites remains a growing issue [13,14].

Virtual walkthrough [15] applications allow viewers to explore a specific place of interest modeled as a virtual environment, containing a vast number of virtual objects, without the need for physical travel [16]. Such applications have demonstrated their effectiveness in enhancing knowledge of tangible cultural heritage and historic places [17,18,19] but also identified challenges for implementation, such as model optimization [20], navigation, and interaction system [18]. Because exploration of virtual environments is not straightforward [21], the navigational interface must be easy to learn, easy to use, and provide good performance. A VR navigation system should strike a balance of immersion and flow, ease of use and mastering, competence and sense of effectiveness, and psycho-physical discomfort, adopting a user-centered design approach and qualitative evaluation methods to assess the user experience [22]. As mouse and keyboard reduce the naturalness of the interaction [23], novel human–computer interaction (HCI) techniques were developed to improve scene visualization and interaction in XR applications, but their use often requires expertise and/or costly devices like smart glasses, head-mounted displays, and their controllers. Freehand locomotion outperforms traditional controller-based locomotion in terms of immersion and task performance [24,25]. Hand gestures serve as a natural and intuitive interaction modality for a variety of application [26,27] and Hand Gesture Recognition (HGR) systems can be used with a simple Red, Green, and Blue wavelengths (RGB) camera [28]; thus, HGR makes VR and XR technology more accessible for collaborative purposes [23,26,28].

This study develops and tests a virtual walkthrough prototyping application for collaborative exploration and learning purposes of an alley within the Sassi of Matera, a UNESCO World Heritage Site in Italy. It combines photogrammetric reconstruction and HGR to give the virtual visitors an interactive and engaging learning experience. Hand gestures are used to navigate through the 3D scene replicating the real-world scenario and interact with virtual dashboards displaying educational content, according to the didactic needs.

The study aims to contribute to the growing body of research on the use of XR technologies for cultural heritage and education and explore the potential of integrating different techniques to create more immersive and engaging virtual walkthroughs that facilitate access to cultural heritage sites and support remote learning activities. The specific scope is to provide users with an integrated and multidimensional educational tool that supports their visit experience prior to the physical visit or during remote learning activities and can be easily accessed. Through the case study, this research demonstrates the potential of such technologies to create a rich and authentic experience that is not limited by physical barriers, allowing for greater accessibility and valorization of sites of historical, archaeological, and architectural interest.

## 2. Materials and Methods

### 2.1. Virtual Reconstruction of an Alley in Matera’s Sassi

The case study was conducted in Matera, a city in the region of Basilicata, Italy. This location was selected due to its unique and culturally rich history [29,30].

The city is renowned for its ancient urban center, formed from the *Civita*, the *Sasso Caveoso*, and the *Sasso Barisano*, and the overlooking *Gravina* canyon, whose geomorphology influenced its development [31]. The Civita is the central spur of the Sassi, surrounded by two torrential incisions (*grabiglioni*) that shaped the housing structures of the Sasso Barisano and Sasso Caveoso built on overlapping terraces. The Sassi, meaning *stones*, form a complex urban landscape carved and constructed from calcarenite (a light-colored calcareous sedimentary rock with a granular texture and organic inclusions) and arranged in a vertical succession of levels that incorporate natural terraces and cuts. The streets often pass above the roofs of other homes. The urban ecosystem [32] is known for its ability to adapt and regenerate, balancing environmental, energetic, and cultural sustainability. In the late 19th and early 20th centuries, the population growth disrupted this balance. In response to the unsanitary conditions, the Risanamento process of socio-economic and urban regeneration was implemented in the latter half of the 20th century. In the late 1980s, the Sassi complex was rediscovered and restored, and in 1993, UNESCO granted it World Heritage status. In 2019, Matera was designated the European Capital of Culture and still showcases a remarkable landscape with unparalleled morphological features.

The location provided a challenging and complex urban environment for the virtual reconstruction process, which aimed to recreate the intricate details of the site with accuracy and precision. The irregularity of the structures, combined with the presence of overhanging elements, hollows, and caves, creates a complex three-dimensional geometry that is difficult to accurately reconstruct through manual 3D modeling [33] and requires the use of advanced imaging techniques, such as photogrammetry [8,9,10,13,20]. The presence of bright and reflective surfaces, such as light calcarenite and glass, further complicates the process as it causes reflections that can distort the appearance of the structures and potentially result in inaccurate capture.

Thus, the reconstruction of a portion of the urban environment was carried out using photogrammetry with the software RealityCapture (©Epic Games, Inc., Cary, NC, USA). This software was chosen because it can process images taken from multiple viewpoints to generate a 3D representation of an object or scene to be used in various applications such as architectural and industrial design, video production [34], games [35], and cultural heritage preservation [8,20].

A total of 1937 photographs (Figure 1) of an alley in the Sassi of Matera were taken with a general-purpose camera (a Nikon D5200 CMOS camera, Torino, Italy) in .jpg format with a resolution of 6000×4000 pixels and a focal distance of 18 mm and then used as input in RealityCapture to generate a highly detailed and accurate 3D representation of the urban scene. The .jpg format was chosen over RAW to reduce device dependency, despite RAW’s potential to alleviate uneven lighting issues.

The output generated by the software was a point cloud (Figure 2), which was used to generate a triangle mesh, a mathematical representation of the 3D objects composed of a set of interconnected triangles.

The reconstructed mesh was highly complex, with a total of 528.1 million (528,104,628) triangles and 264.7 million (264,716,302) vertices. This level of complexity tends to create errors (e.g., out of memory and other runtime errors) and could lead to longer processing times, larger file sizes, and reduced scalability. Furthermore, the photographs were taken outdoors on a sunny summer day, so some of them presented many reflections and were very bright. RealityCapture software struggled to handle reflective and transparent objects, resulting in deformations and missing parts in the 3D reconstructed model. Therefore, it was necessary to simplify the mesh by reducing the number of triangles to approximately 9 million (8,974,781) and removing those parts that were unnecessary and degraded due to reflections of light; light-colored objects such as plastic tables and chairs in the alley showed degraded shapes and many inconsistencies in the topology of the reconstruction. These two processes were achieved directly within RealityCapture, utilizing the *Simplify* and *Filter Selection* tools. The first tool reduced the triangle count of the selected reconstructed mesh by specifying the target triangle count while preserving its overall shape and structure. The latter tool created a new model from the previous one by removing specified triangles that were manually selected by basic modes (e.g., lasso, rectangular, and box selection) and via the *Advanced Selection* tool by different criteria (e.g., marginal, large, small, and largest connected). The simplification process helped to balance the trade-off between the desired level of detail and performance.

To improve the photo-realism, a total of 6 high-quality textures with a resolution of 16,384×16,384 pixels were then created based on the complex mesh and reprojected onto the simplified model, using the *Texture Reprojection* tool of RealityCapture, which preserves the illusion of a high-quality reconstruction with a mesh of minimal triangle count. The final textures, generated by aligning the input photos and projecting them onto the 3D model, were reduced to a total of 5 with a resolution of 4096×4096 pixels.

The figures that follow provide a detailed visualization of the reconstructed mesh within the RealityCapture user interface. Figure 3 presents the reconstructed mesh without any texture applied, providing a clear view of the geometric structure of the model.

Figure 4 presents the same view of the mesh but with the texture obtained from the input photos applied, resulting in a more realistic representation of the scene. Upon closer inspection, it can be observed that the shadows of the removed objects are still visible in the texture of the filtered mesh, as can be seen in the bottom right corner of Figure 4.

The comparison presented in Figure 5 offers a closer look at the effects of the filtering process on the reconstructed 3D model. The zoomed-in view of the alley showcases the mesh texture with and without the deformed objects, i.e., before and after filtering. Despite the filtering process, the shadows of the removed objects are still noticeable.

The reconstructed model was subsequently exported in the fbx format and imported into a project made with Unreal Engine (UE) 4.27 (©Epic Games, Inc., Cary, NC, USA) [36], which had been previously designed for multiplayer virtual reality experiences, using a template [37]. Figure 6 shows the editor interface with the scene reconstruction.

The lighting strategy adopted in UE aimed to prevent incorrect rendering of lights and shadows. This was achieved by exclusively illuminating the scene with the SkyLight Object, which serves as an ambient light source. In addition, the textures were utilized as *Emissive Color* in the materials of the object.

The 3D model obtained through the workflow described earlier served as a virtual replica of the real-world alley to be used as the environment for the virtual walkthrough’s prototypical application.

### 2.2. HGR System

The HGR system used in this study was adapted from the work presented in [28]. Figure 7 depicts the subsequent stages of the pipeline:1.Input capture: a frame is acquired via a general-purpose monocular RGB camera;2.MediaPipe Hand [38,39] processing: The MediaPipe Palm Detection Model and the Hand Landmark Model are applied to the captured frame to detect hands and predict hand–knuckle coordinates.3.Hand gesture prediction: A Feed-Forward Neural Network (FFNN) is used to predict the current hand gesture from the *x* and *y* components of each hand–knuckle coordinate.

The use of a general-purpose monocular RGB camera as an input source has been shown to achieve good hand tracking [40]. Additionally, RGB cameras are widely available and relatively inexpensive, making them a cost-effective solution for many applications, as they can be found onboard on most laptops and smartphones.

MediaPipe Hands is a well-established approach due to its reliance on single camera-independent RGB frames as input, in combination with its real-time inference speed and high prediction quality [28,38,39]. Furthermore, MediaPipe Hands is a free and open-source solution that can be customized and used across different platforms.

MediaPipe Hands [38,39] uses two Deep Learning (DL) models trained on both real-world and synthetic images: the Palm Detection Model, which returns an oriented hand bounding box, and the Hand Landmark Model, which returns hand keypoints. The Palm Detection Model analyzes the full image and identifies hand bounding boxes by employing strategies such as detecting the palm instead of the hand to address self-occlusion and complexity issues, using square bounding boxes to reduce anchors, applying an encoder–decoder feature extractor to improve context awareness, and minimizing focal loss during training to handle high scale variance. These techniques achieve an average precision of 95.7%. The subsequent Hand Landmark Model locates keypoints of 21 hand–knuckle coordinates within the boundaries of detected hand regions through regression. It learns a consistent internal representation of hand poses and performs well even with partial visibility and self-occlusions. These coordinates consist of *x*, *y*, and *z* components, where the *z* component is obtained using relative depth with respect to the wrist, and for this reason, the coordinate spaces are referred to as 2.5D [38].

The FFNN is provided with 2D coordinates (*x*, *y*), defined in the pixel space, of each of the 21 predicted landmarks as input. For the specific task of interest, the third coordinate *z* was not informative and thus could be disregarded.

A dictionary of 13 hand gestures was defined based on their potential use in an interactive virtual urban environment. These gestures, derived from state-of-the-art research [41,42,43,44], constitute a subset of the 15 gestures proposed in [28] and are represented by various combinations of open and closed fingers and hands.

The proposed gesture set is divided into two categories: 6 static gestures (thumb up, thumb down, open hand, ok, peace, rock) and 7 dynamic gestures (simple pinch, combo pinch, simple rotation, combo rotation, two fingers swipe right/left, three fingers swipe right/left, four fingers swipe right/left), shown in Figure 8.

Static gestures are fixed hand poses predicted directly by the FFNN. The FFNN inputs the coordinates provided by the MediaPipe framework, based on the current camera frame. The system considers a gesture valid if it outputs the corresponding gesture representation.

Dynamic gestures are defined by hand movement and further divided into single and combo gestures depending on the number of hands tracked. Single dynamic gestures are performed by a single hand, whereas combo dynamic gestures are performed by two hands. The system uses static activation gestures, predicted by the FFNN in a similar manner to static gestures, to initiate the tracking of dynamic gestures. It then employs assertions on the landmarks over subsequent frames to determine the type and direction of the dynamic gesture performed by the user.

The distinction between static and single or combo dynamic gestures allows for a more nuanced understanding of the user’s intention and can lead to a more intuitive interaction experience.

The FFNN architecture features a single hidden layer with 32 units and employs a rectified linear unit (ReLU) activation function on the hidden layer and a Softmax function on the final layer. The choice of the FFNN among other DL approaches and its implementation strategies were carefully selected to achieve an optimal balance between computational cost, accuracy, and real-time performance, as detailed in [28].

To train the network, a dataset of 130,000 hand pose samples was used. The samples were taken with different hands and cameras and were manually labeled with a balanced distribution among gestures. The dataset was divided into a training and a validation set in an 80:20 ratio. After training, the network was tested in real-time using different devices: an Intel RealSense D455 camera, a 40MP Huawei P30 back camera, and a 1080p MacBook Pro (M1 Pro) camera. Each sample consists of 42 elements, including the *x* and *y* values of 21 hand landmarks, provided by the MediaPipe processing step. As MediaPipe also supports egocentric hand tracking, the samples were taken and labeled in both egocentric and non-egocentric modes, allowing the system to be used with a simple RGB camera placed in front of the user (e.g., laptop webcams) or mounted on XR devices. The FFNN was trained for 2000 epochs using the Adam optimizer [45] algorithm with a learning rate of 0.0001. The training resulted in a prediction accuracy of 98% [28].

### 2.3. Collaborative Virtual Walkthrough Application

A virtual walkthrough prototyping application of the alley within Matera’s Sassi was developed for collaborative exploration and learning purposes. It combines the photogrammetric reconstruction and the HGR system to give the virtual visitors an interactive and engaging learning experience. The virtual environment was augmented into the UE project with the capability of supporting interactions and movements within the virtual space through the HGR system. Hand gestures are used to navigate through the 3D scene replicating the real-world scenario and interact with virtual dashboards displaying educational content. Table 1 shows the dictionary created to associate hand gestures with navigation actions in the virtual space. The mapping of gestures to actions was designed with the aim of ensuring that the system is intuitive and easy to use for non-expert users; multiple gestures for the same action provide users with flexibility and reduced likelihood of mistaken gestures.

In selected points within the reconstructed alley, as shown in Figure 9, virtual 3D information stands were incorporated to showcase virtual dashboards reporting a combination of real-time data and static information pertaining to cultural heritage and environment, such as narratives on diachronic evolution of urban morphology, opportunistic monitoring data [46,47], and images of the local fauna and flora.

The dashboards—accessible through the mentioned 3D information stands placed in the scene—were triggered by a pre-defined gesture as popup windows (refer to Table 1). The information can be customized to suit the didactic needs and the desired storytelling.

The above-described steps were performed on a high-performance computer workstation equipped with a Microsoft Windows 10 Pro 64-bit operating system, an Intel Xeon W 2275 (Cascade Lake-W) processor with a clock speed of 3.30 GHz, 256 GB of RAM, and a NVIDIA RTX A6000 graphics processing unit (GPU). This setup ensured optimal performance and stability for the tasks at hand.

## 3. Results

The immersive virtual walkthrough application developed in this study combined photogrammetric reconstruction and a deep learning-based HGR system to offer a highly interactive and collaborative experience for multiple users to explore the historical and cultural site of Matera’s Sassi. The virtual environment created by the application provided precise representations of Matera’s complex historical locations, enabling users to interact with the virtual environment using natural hand movements recognized by the HGR system. The application included educational content on cultural heritage and the environment, incorporating real-time data, static information, and visual media.

Figure 10 and Figure 11 depict the same views as the photographs in Figure 1, captured within the running application.

The application was tested on a sample group of 36 participants who were representative of both genders and composed of high school students. The participants were asked to perform various tasks within the virtual environment, such as exploring the alley, interacting with the virtual dashboards, and navigating the space using the gestures defined in Table 1. The aim of this test was to assess qualitative aspects [22] of the overall user experience of the application.

The subjective feedback was collected using questionnaires that asked participants to rate their experience on several parameters, such as level of immersion, ease of use, level of engagement with the educational content, overall satisfaction, and recommendation to others. When rating their experience, participants used a scale of 1 to 5, with 5 representing the highest score. The results of the questionnaires showed that the majority of participants found the application easy to use and engaging. The use of hand gestures for navigation and interaction was praised for being intuitive and enhancing the sense of presence in the virtual environment. The average score for the ease of use was 4.50, while the average score for the level of immersion was 4.11. Additionally, the majority of participants reported a positive experience with the application (more than 88%), expressed engagement with the educational content (83%), and stated that they would recommend it to others (86%). The integration of educational content on cultural heritage and environment was also positively evaluated, with participants highlighting the educational value of the virtual experience.

The results of the questionnaires are illustrated in Figure 12.

The study did not include task completion time, as it prioritized the users’ experiential and qualitative aspects over performance measures. This allowed users to freely explore the environment at their own pace and in their preferred way.

These results indicate that the virtual walkthrough application for collaborative exploration and learning was well-received by the participants. The participants found the HGR easy to use and the created virtual environment detailed, representing accurately the real-world location. The HGR accurately recognized in real time a variety of hand gestures with inputs provided by a general-purpose monocular RGB camera (i.e., a consumer webcam) and translated them into actions in the virtual environment. Our findings showed that having multiple gestures for the same action in a gesture-based navigation system can enhance user experience and accessibility; the approach enables users to choose the most comfortable and instinctive gesture, can provide redundancy in case of incorrect recognition, and can reduce the likelihood of mistaken gestures. The students were able to explore the virtual environment together and interact with virtual objects and elements in a natural and intuitive way, improving the learning experience, with many expressing a desire for continued use. However, it is important to acknowledge that the sample size of the study was limited, and further research with a larger and more diverse participant group would be beneficial to confirm the generalizability of these findings.

## 4. Discussion and Conclusions

Overall, this study provides valuable insights and demonstrates the potential of virtual walkthrough applications using photogrammetric reconstruction and HGR to enhance the user experience in collaborative virtual learning environments.

The aim is to provide users with an integrated and multidimensional educational tool that supports their visit experience prior to the visit or during remote learning activities (if it is impossible to physically visit the location) and can be easily accessed.

The findings indicate that this approach is effective in promoting the cultural landscape of Matera, which encompasses various aspects such as archaeological, historical, and natural heritage. The immersive nature of virtual environments can offer an engaging and accessible learning approach, and incorporating hand gestures can enhance the experience’s intuitiveness and user-friendliness.

The user test led to positive feedback, highlighting the advantages of integrating advanced techniques (e.g., DL-based HGR systems, photogrammetry, collaborative tools) to provide precise representations of complex historical real-world locations as well as additional narratives in immersive applications for architecture, cultural heritage, and environmental education. The proposed application has the potential to promote the cultural landscape of Matera and be applied in various different domains, including but not limited to preservation, gaming, and tourism [48].

Future work should aim at exploring new ways to make the virtual experience even more realistic, as well as improving the performance and scalability of the system. This could involve expanding virtual walkthroughs to include additional historical and cultural sites. Additionally, further studies may explore the use of other forms of input, such as voice recognition, to enhance the experience even further. However, deeper analysis is necessary to investigate the impact of these technologies on learning outcomes and user experience; qualitative and quantitative assessments [49], such as task completion time, users’ stress responses, and cognitive load [12], could provide a more comprehensive evaluation to fully understand the benefits and limitations of these kinds of applications.

## Figures and Tables

**Figure 1 jimaging-09-00088-f001:**
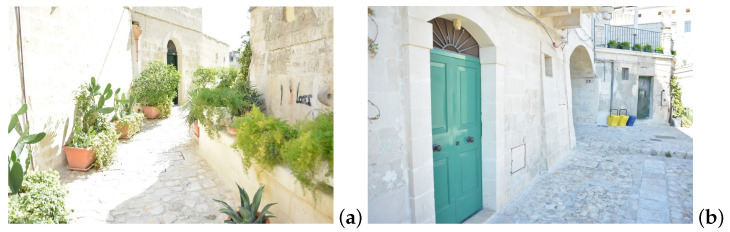
Two examples of the 1937 photographs used as input in RealityCapture software for the 3D reconstruction of an alley in Sassi of Matera. (**a**) A perspective from the entrance of the alley. (**b**) A perspective from the end of the alley.

**Figure 2 jimaging-09-00088-f002:**
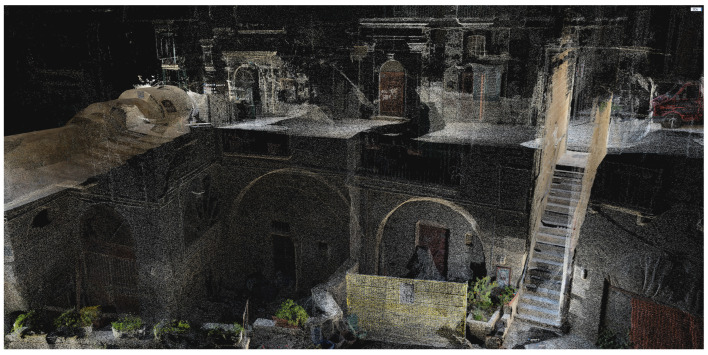
A portion of the 3D point cloud of the alley, depicting the morphological and material complexity typical of the Matera’s Sassi urban landscape.

**Figure 3 jimaging-09-00088-f003:**
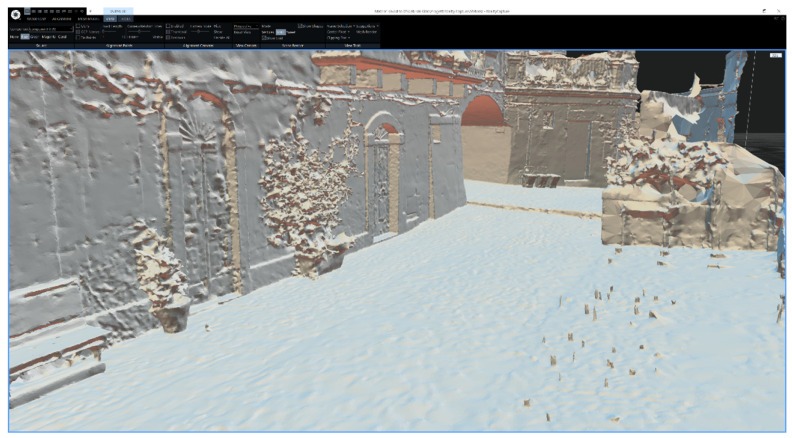
RealityCapture user interface showing the reconstructed mesh.

**Figure 4 jimaging-09-00088-f004:**
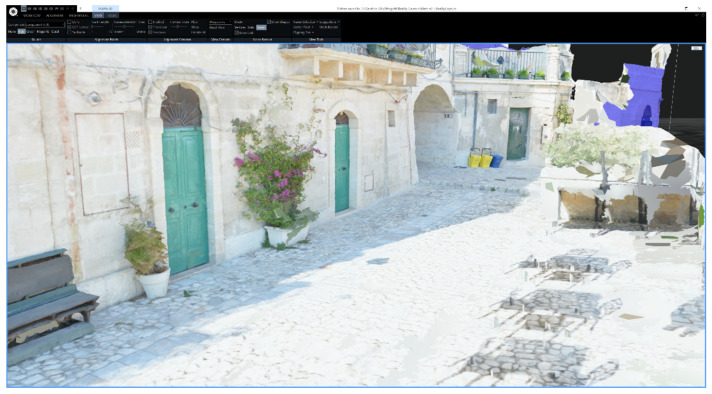
RealityCapture user interface showing the reconstructed mesh with the texture applied.

**Figure 5 jimaging-09-00088-f005:**
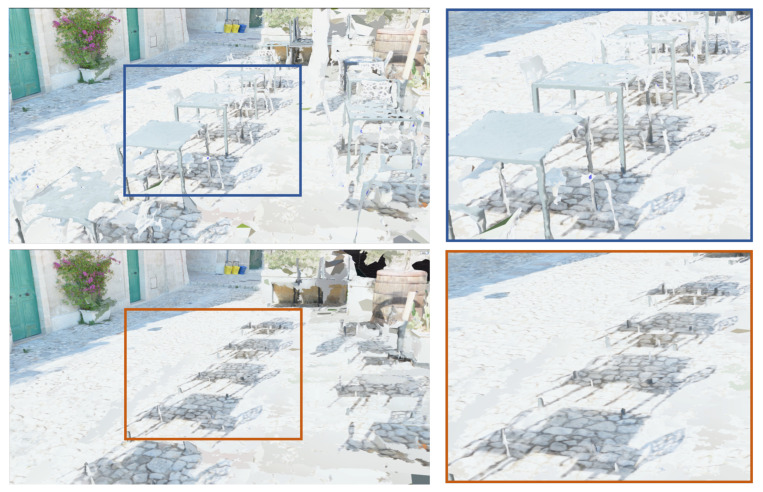
Before and after the removing of the deformed objects, with a zoomed-in view where their shadows are still observable.

**Figure 6 jimaging-09-00088-f006:**
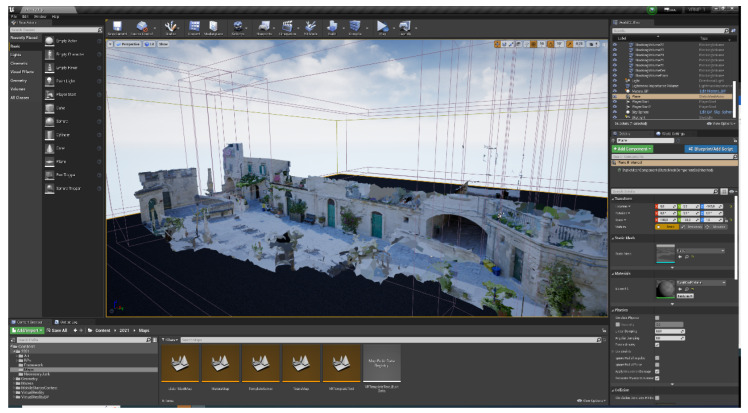
Unreal Engine (UE) user interface showing the virtual scene.

**Figure 7 jimaging-09-00088-f007:**
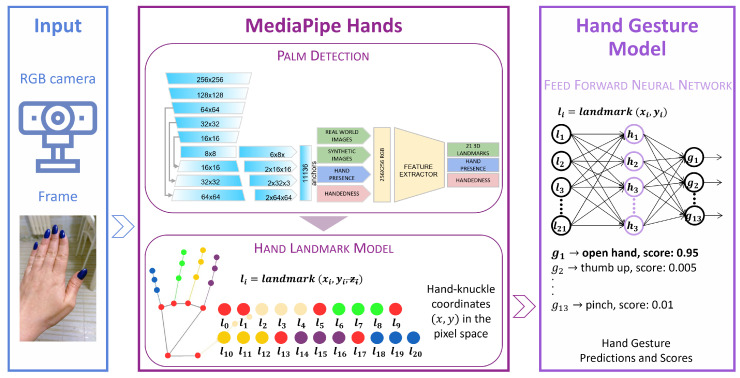
HGR System pipeline. Reproduced from [28].

**Figure 8 jimaging-09-00088-f008:**
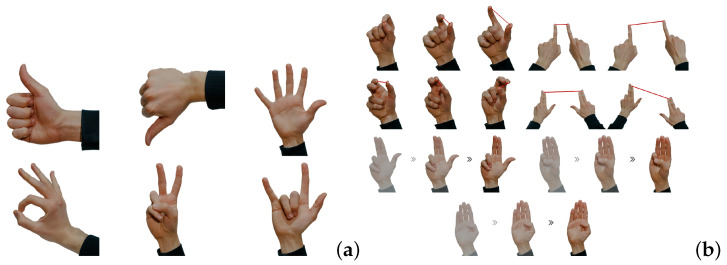
Proposed gesture set. (**a**) Static gestures: thumb up, thumb down, open hand, ok, peace, rock. (**b**) Dynamic gestures: simple pinch, combo pinch, simple rotation, combo rotation, two fingers swipe right/left, three fingers swipe right/left, four fingers swipe right/left. Adapted from [28].

**Figure 9 jimaging-09-00088-f009:**
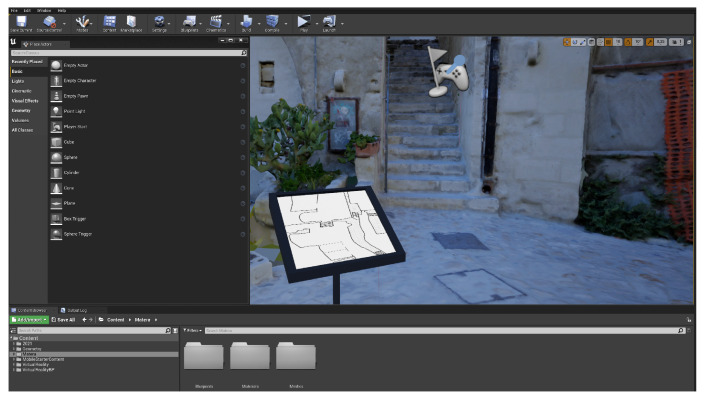
UE editor view of a customizable virtual 3D information stand placed within the reconstructed alley.

**Figure 10 jimaging-09-00088-f010:**
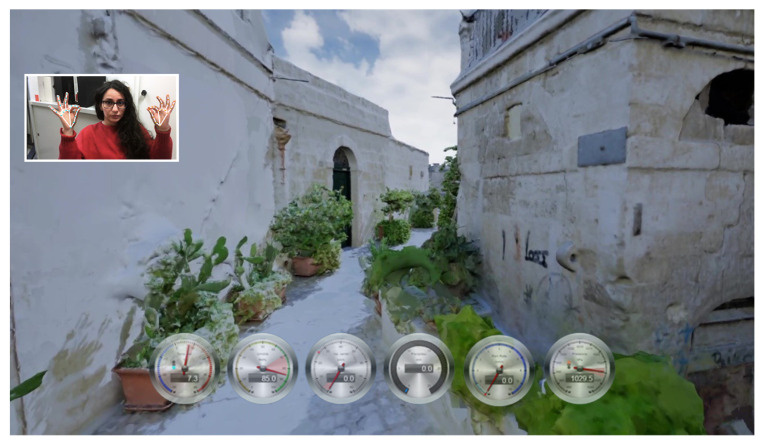
A view of the virtual walkthrough application, with a real-time virtual dashboard.

**Figure 11 jimaging-09-00088-f011:**
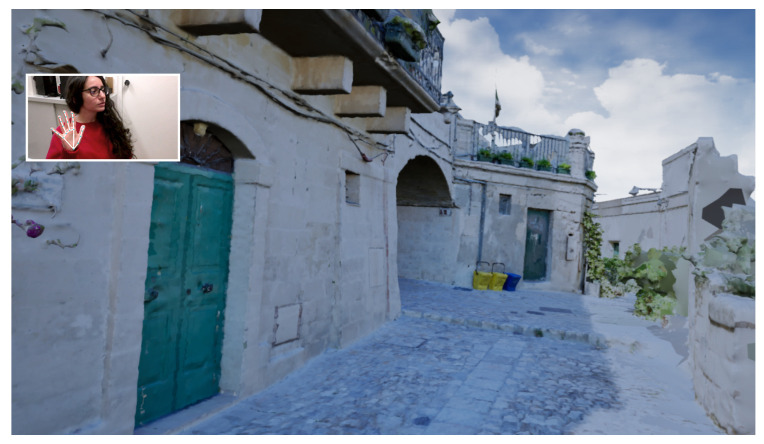
A view of the virtual walkthrough application during navigation.

**Figure 12 jimaging-09-00088-f012:**
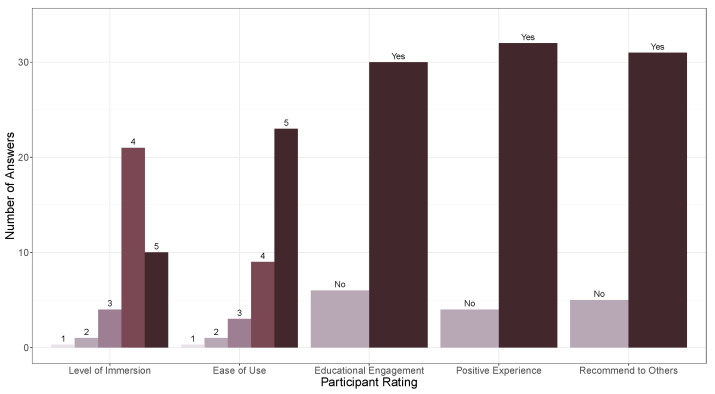
Results of questionnaire-based subjective feedback from 36 high school student participants on the virtual environment application, including ratings on level of immersion, ease of use, level of engagement with the educational content, overall satisfaction, and recommendation to others.

**Table 1 jimaging-09-00088-t001:** Gestures and associated actions.

Gesture	Action	Type
Thumb up	Move forward	Static
Thumb down	Move backward	
Open hand	Stop	
Ok	Teleport forward	
Peace	Teleport backward	
Rock	Jump	
Simple pinch	Open/close popup	Dynamic
Combo pinch	Open/close popup	
Simple rotation	Look up/down	
Combo rotation	Look up/down	
Two fingers swipe right/left	Look right/left	
Three fingers swipe right/left	Look right/left	
Four fingers swipe right/left	Look right/left	

## Data Availability

The study’s data are available upon request from the corresponding authors for academic research and non-commercial purposes only. Restrictions apply to derivative images and models trained using the data, and proper referencing is required.

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
