# Peer review of "A Collaborative Virtual Walkthrough of Matera’s Sassi Using Photogrammetric Reconstruction and Hand Gesture Navigation"

_2313-433X, 2023, doi:10.3390/jimaging9040088_

Round 1

Reviewer 1 Report

I think the total of 49 references is too large. Multiple papers illustrating the application of the techniques to other situations are not necessary (e. g. [33] and [34]), and I also believe that some of the citations in the Introduction could be removed.

Line 87 on page 2: The words “lying in their ability” are not needed for the meaning or grammar of the sentence and should be eliminated.

Lines 153 and 154 on page 4: replace “let to preserve” by “preserves”.

Line 209 on page 7. I do not understand the meaning of the words “using relative depth with respect to the wrist”. If this means “keypoint depth minus depth of the center of the wrist” then I would regard it as a fully 3D depth coordinate. To me, 2.5 D means that the depth is given as membership in one of a small number of discrete layers.

Page 8: Figure 8 is a combination of figures 2 and 3 of [27], which is not appropriate, because that paper considered 8 static gestures, while this one uses only 6. I recommend cutting and pasting only the 6 gestures used in this paper into figure 8 part a). Otherwise, the caption to that figure part a) is confusing. Also, please list the 6 static gesture names in the caption in the order that they appear in the figure. If I am interpreting those gestures correctly, “rock” appears before “ok” in the figure but is listed after “ok” in the caption.

I was confused by table 1 on page 9 because several dynamic gestures were matched to the same action meaning. This was explained later on lines 315 – 319, but that was too late to prevent my confusion. It should be explained at line 260 when that table is first referenced.

I appreciated the list of abbreviations because it is potentially helpful. But it should appear on the first or second page, so that readers know it exists and can use it. If it appears at the very end, many readers will not know it is there until too late, after they have read the whole paper. That is what happened to me. I had forgotten the meaning of UE and had to start searching from the beginning through the paper for its definition. When finally I saw the table, I wished I had known about it earlier.

Reviewer 2 Report

A very good introduction with a brief and on-point synthesis of the developments in the field. The motivation for the chosen approach is well-defined.

Minor suggestions in this part:

"real-time visualization of reality-based 3D surveying and modeling of heritage sites, remains a growing issue [13]" - the statement is valid but I would suggest adding a more recent reference for better support of it. 

"Virtual walkthroughs [14] application allows viewers (...)" - perhaps "virtual walkthrough [14] applications" or "a virtual walkthrough [14] application" would work better for the sense of the phrase.

--

The case study subject is well presented. The reasoning for the choice of photogrammetry as the digitization method is good. The photogrammetric workflow and the 3D processing are briefly presented.

The two images shown as examples from the photogrammetric image set showed the problems explained later on about the difficulty in the 3D processing due to the high reflection/shadows/highlights. The method used to solve/avoid these problems is well described but expert readers will raise the question of why not shooting in RAW instead of .jpg in order to alleviate some of these issues. Perhaps a short explanation about this would help, but given the main subject of the paper, is not absolutely necessary. 

Showing the mesh surface without texture and the problems resulting from the working conditions during the photo shooting (sunny day) is appreciated as transparency of the workflow. Also, I would suggest it is a good opportunity to explain what would be some advice in order to avoid or alleviate the encountered problems (like shooting RAW, choosing an overcast day, removing shadows/de-lighting the images, etc.). 

Also while talking about the optimization of 3D models using decimation/simplification it would be worth mentioning the use (if it is the case) of Normal maps (generated from the high-quality mesh) along with the color textures. If Normal maps were not used, I recommend it to improve the light interaction perception with the 3d model surface in the 3d viewer.

---

Although I am not familiar with HGR systems, the proposed method is well presented and the algorithm choice is well argued. The system workflow, parameters, and training are very well described.

---

The Walkthrough Application development is briefly described, giving more emphasis to its features rather than how they were implemented. 

The results are well presented. The survey contributes to the validation of the approach although, as the authors mention, a larger number of participants and a few detailed technical-oriented questions would have helped for even more precise feedback and further improvements to the app. 
